# A molecular view on the escape of lipoplexed DNA from the endosome

**Bart MH Bruininks, Paulo CT Souza, Helgi Ingolfsson, Siewert J Marrink***

Groningen Biomolecular Sciences and Biotechnology Institute and Zernike Institute for Advanced Materials, University of Groningen, Nijenborgh, Netherlands

**Abstract** The use of non-viral vectors for in vivo gene therapy could drastically increase safety, whilst reducing the cost of preparing the vectors. A promising approach to non-viral vectors makes use of DNA/cationic liposome complexes (lipoplexes) to deliver the genetic material. Here we use coarse-grained molecular dynamics simulations to investigate the molecular mechanism underlying efficient DNA transfer from lipoplexes. Our computational fusion experiments of lipoplexes with endosomal membrane models show two distinct modes of transfection: parallel and perpendicular. In the parallel fusion pathway, DNA aligns with the membrane surface, showing very quick release of genetic material shortly after the initial fusion pore is formed. The perpendicular pathway also leads to transfection, but release is slower. We further show that the composition and size of the lipoplex, as well as the lipid composition of the endosomal membrane, have a significant impact on fusion efficiency in our models.

***For correspondence:**
s.j.marrink@rug.nl

**Competing interests:** The authors declare that no competing interests exist.

## Introduction

Gene therapy is a promising technique with a wide applicability. The first clinical trials with gene therapy started in the early 90's, and the first approved therapy being introduced in Europe in 2012 (*Cressey, 2012*; *Blaese et al., 1995*; *Hanna et al., 2017*). Even though the concept of gene therapy has been around for a while, the problem remains to target and enter the right cells, without being toxic to the rest of the organism.

Most higher organisms have evolved quite stringent measures to block uptake of DNA from their surroundings, preventing excessive genetic instability. Naked DNA gets quickly degraded in our body by exonucleases, whereas using viruses as vectors can lead to a strong acquired immune response (*Nayak and Herzog, 2010*). Therefore, 'new' non-viral based vectors are being developed. Non-viral vectors have two advantages, first, they do not trigger a specific immune response, and second, they are potentially much cheaper than viral vectors. Most non-viral vectors use cationic lipids or polymers for complexation with the negatively charged DNA, concealing the genetic material from degradation. The major downside to many of these non-viral based methods, is that their transfection rates in vivo are rather low (*Tros de Ilarduya et al., 2010*; *Rezaee et al., 2016*; *Nayerossadat et al., 2012*). Moreover, lipoplexes are potentially toxic especially if they are highly positively charged (*Huang and Li, 1997*).

In this work we focus on cationic lipid-DNA based vectors, called lipoplexes. Lipoplexes consist of (cationic) lipids whose role is twofold. One, they shield the genetic material from degradation, and two, the lipids support transfection (*Kim et al., 2015*; *Ciani et al., 2004*; *Felgner et al., 1987*). Upon exposure of the vector to cells, the cells incorporate the lipoplex by means of endocytosis, causing the complex to subsequently reside in the early and late endosome, and finally to be degraded in the lysosome. It is therefore important for the genetic material to escape the endosome before it gets degraded (*ur Rehman et al., 2013*; *Rejman et al., 2004*; *Zuhorn et al., 2007*). This escape from the endosome is one of the least understood and inefficient steps in current lipoplex or

polymer mediated gene therapy and improving endosomal escape might improve transfection efficiency drastically (*Degors et al., 2019*).

Due to the microscopic scale and dynamic nature of lipoplex-membrane fusion, it has proven very difficult by experimental means to relate physico-chemical properties of the vector to transfection efficiency. Molecular dynamics (MD) simulations provide an alternative tool to study molecular processes in atomic, or near atomic detail (*Marrink et al., 2019*). In particular coarse-grained (CG) models such as Martini (*Marrink et al., 2007*; *Uusitalo et al., 2015*) have proven popular, trading some of the atomic detail for a computational speedup and enabling direct simulation of membrane fusion (*Markvoort and Marrink, 2011*; *Smirnova et al., 2019*; *Pannuzzo et al., 2014*). Here we use the Martini model to simulate the fusion between nanoscale lipoplexes and endosomal model membranes and vesicles. We are able to resolve the molecular details of the fusion process, including the full release of the lipoplex cargo, short fragments of double-stranded DNA (dsDNA), across the endosomal membrane.

## Results and discussion

### Construction and validation of the lipoplex model

To investigate the escape of genetic material from a lipoplex inside a model endosome, we first constructed and validated a small lipoplex (~18 nm in diameter, 4 fragments of 24 bp dsDNA; *Figure 1—figure supplement 1*). Our lipoplex formulation consists of 1,2-dioleoyl-3-trimethylammonium-propane (DOTAP) as the cationic lipid to bind the dsDNA, and 1,2-dioleoyl-sn-glycero-3- phosphoethanolamine (DOPE) as the helper lipid, at a 1:4 DOTAP/DOPE ratio. This complex is known to form the inverted hexagonal phase ($H_{II}$) in complex with dsDNA at this ratio (*Kim et al., 2015*; *Corsi et al., 2010*; *Koltover et al., 1998*). Starting from a multilamellar initial configuration, with the dsDNA sandwiched in between the lamella, we observe spontaneous formation of the inverted hexagonal phase (*Figure 1—figure supplement 1*), consistent with the experimental behavior. After removing periodic boundary conditions and coating the aggregate with an additional layer of DOTAP and DOPE, we arrive at our small lipoplex model which proves stable during a 10 µs simulation (*Figure 1I*; *Figure 1—figure supplements 1* and *2*). To validate our model, we compared the dsDNA spacing inside the solvated lipoplex with the available SAXS data from literature. This showed that the liquid crystal lattice of the inverted hexagonal phase in our solvated CG lipoplex was very close to the experimental data (6.0 ± 0.1 nm compared to 6.1 nm for SAXS) (*Corsi et al., 2010*).

For the endosome we made a crude symmetrical membrane model containing a mixture of PC and PS lipids at a 4:1 ratio. Clearly, real endosomal membranes are much more complex concerning lipid composition (*van Meer et al., 2008*). However, to a large extent, the exact composition is unknown (especially concerning asymmetry) and varies between endosomal stage and cell type. Given the large complexity of the simulated systems, we decided to keep the endosomal membrane composition as simple as possible, with PC as main phospholipid. PS was added as it was hypothesized before that anionic lipids could play an important role in the transfection mechanism (*Tarahovsky et al., 2004*). Both PC and PS lipids had a mixture of diC14:0 (dimyristoyl), diC14:1,9 c (dimyristoleoyl) and C14:0-C14:1,9 c (myristoyl-myristoleoyl) tails at a 1:1:2 ratio (*Figure 2*). The relatively short lipid tails were chosen because pore formation for longer tailed lipids have been shown to be too energetically unfavourable for the CG Martini lipids compared to their all-atom counterparts (*Cohen and Melikyan, 2004*; *Bennett and Tieleman, 2011*).

### Two distinct mechanisms leading to gene transfection

We first performed completely unbiased fusion experiments by placing the solvated lipoplex in solution above a hydrated bilayer. To monitor fusion we used: (i) lipid mixing, (ii) DNA-water contacts, and (iii) visual inspection. At the 10 µs time scale using unbiased simulations, we did not obtain sufficient dehydration of the lipoplex membrane interface for fusion to proceed (data not shown). Dehydration of the membrane-membrane interface is a known kinetic barrier for membrane fusion (*Leikin et al., 1987*), therefore we changed the initial set-up of the fusion experiment and started from a dehydrated state, with one of the corners of the lipoplex pointing toward the endosomal membrane. The dehydrated systems either remained in their dehydrated state, resulting eventually

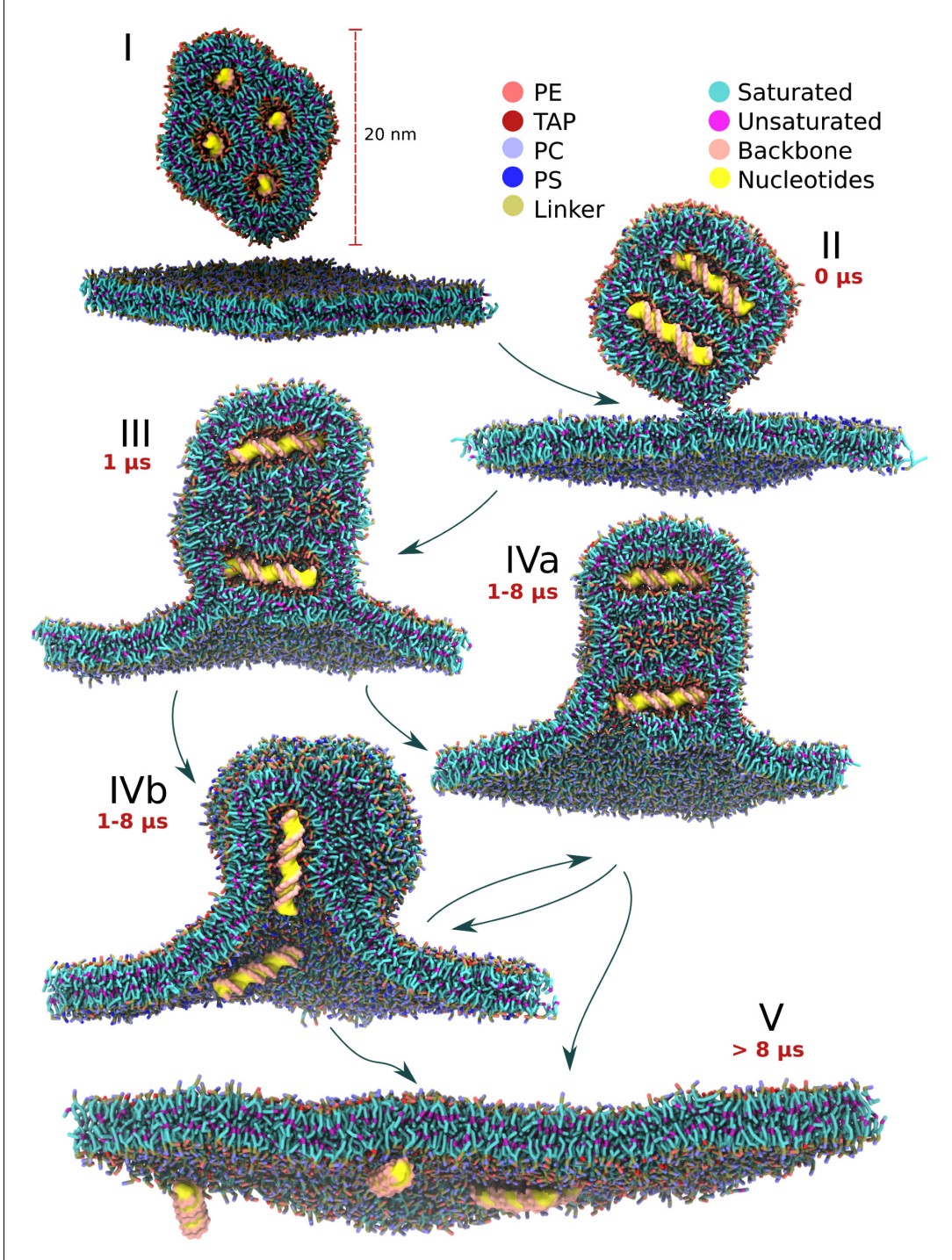

**Figure 1.** Schematic view of the lipoplex-membrane transfection pathways. Unadhered lipoplex containing four dsDNAs sits above the endosomal bilayer (I). After initial stalk formation (II), a wide hemifusion diaphragm is formed (III). A pore is formed in one of the channels containing the DNA (IVa, IVb). The angle of the DNA with respect to the average bilayer normal can be either perpendicular or parallel, resulting in subsequently zipper (IVa) or ejection (IVb) like release of the DNA (V). The pathways indicated are based on 25 independent fusion experiments, each 10 µs long, leading to 34 dsDNA transfection events. The time scales indicated are typical for successful transfection events.

The online version of this article includes the following figure supplement(s) for figure 1:

**Figure supplement 1.** Building the small lipoplex.

**Figure supplement 2.** Radius of gyration for the solvated lipoplex.

**Figure supplement 3.** Coating the lipoplex.

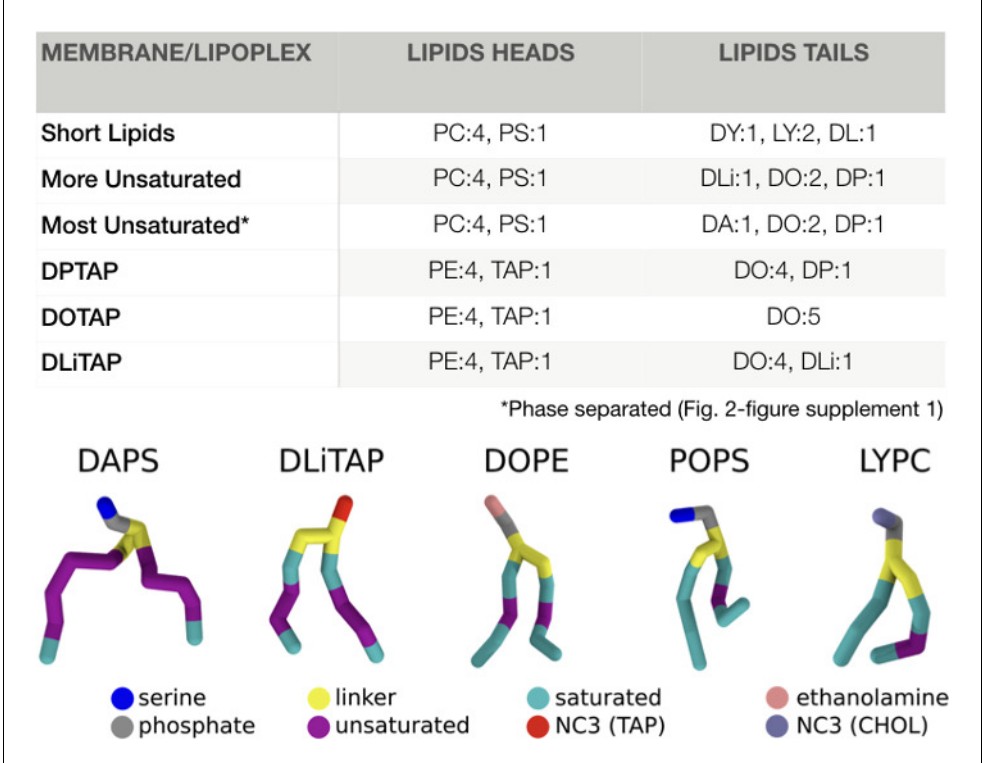

| MEMBRANE/LIPOPLEX | LIPIDS HEADS | LIPIDS TAILS |
|---|---|---|
| Short Lipids | PC:4, PS:1 | DY:1, LY:2, DL:1 |
| More Unsaturated | PC:4, PS:1 | DLi:1, DO:2, DP:1 |
| Most Unsaturated* | PC:4, PS:1 | DA:1, DO:2, DP:1 |
| DPTAP | PE:4, TAP:1 | DO:4, DP:1 |
| DOTAP | PE:4, TAP:1 | DO:5 |
| DLiTAP | PE:4, TAP:1 | DO:4, DLi:1 |

*Phase separated (Fig. 2-figure supplement 1)

DAPS  DLiTAP  DOPE  POPS  LYPC

- serine
- phosphate
- linker
- unsaturated
- saturated
- NC3 (TAP)
- ethanolamine
- NC3 (CHOL)

**Figure 2.** Membrane and lipoplex compositions. The membrane compositions for the fusion experiments, and the lipoplex formulations are indicated in molar ratios. All PC/PS heads were combined with all tails, resulting in six different lipids per membrane composition (first 3). For the lipoplex formulations all PE was linked to DO and all TAP to either DP/DO/DLi (last 3). The nomenclature follows the default Martini lipids abbreviations as described by T.A. *Wassenaar et al. (2015)*. An example of each tail and headgroup is displayed under the table (the tails are in the order of occurrence in their name). The complete nomenclature can also be found online at: 'http://cgmartini.nl/index.php/force-field-parameters/lipids2/350-lipid-details'.

The online version of this article includes the following figure supplement(s) for figure 2:

**Figure supplement 1.** Lipid mixing in all endosome bilayer compositions.

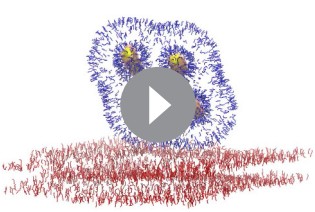

**Video 1.** Small lipoplex-bilayer transfection. Fusion of a small lipoplex containing 4 fragments of dsDNA. The headgroups of the lipoplex lipids (DOTAP:1, DOPE:4) are depicted in blue and the bilayer lipids (short lipids *Figure 2*) are depicted in red. Yellow and pink are used for subsequently the bases and backbone of the dsDNA. No bias was applied during this simulation. https://elifesciences.org/articles/52012#video1

in the formation of a fusion stalk, or the membrane would rehydrate – from which dehydration was never observed. Since the formation of the fusion stalk has been described at many levels of detail (*Knecht and Marrink, 2007*; *Aeffner, 2012*), and is widely accepted as an essential step in any fusion pathway, we decided to start our further transfection experiments from the stalk state (*Figure 1II*), allowing for a more systematic approach.

We then performed five replica simulations, each ten microseconds long, starting from the stalk state. In each case, the system evolved to fusion of the lipoplex with the endosomal membrane, with the cargo being delivered across the target membrane (*Figure 1*; *Video 1*). The initial stalk rapidly decays (on a time scale of 10–100 ns), expanding radially, leading to a so-called hemifusion diaphragm (HD) state (*Kozlovsky et al., 2002*) in which the inner

lipoplex coat and the external endosomal membrane are in direct contact, exchanging lipids (*Figure 1III*; *Figure 3—figure supplements 1–5*). Transfection subsequently proceeds via two different pathways, depending on the orientation of dsDNA either parallel or perpendicular with respect to the membrane normal (*Figure 1IV*; *Figure 3—figure supplements 1–5*). For perpendicular transfection to the bilayer normal, an initial pore is formed at the rim of the hemifusion diaphragm, this causes rapid unzipping of lipids away from one dsDNA fragment. This unzipping results in a pore as large as the dsDNA fragment effectively transfecting the DNA by moving the lipids (*Figure 1IVa*), indicated by a rapid increase in hydration of the dsDNA (*Figure 3—figure supplements 1–5*). For parallel transfection, a pore is formed at the place where the nearest dsDNA fragment is pointing at the HD. Pore formation is then followed by 'ejection' of this dsDNA fragment, resembling the unloading of a syringe, by moving both the dsDNA and lipids (*Figure 1IVb*). Visual inspection of the unloading process (*Video 1*) reveals that the dsDNA moves out of the channel concomitantly to the merging of the surrounding lipids with the endosomal membrane. This suggests that the release of curvature stress of the lipoplex (as the lipoplex channel lipid composition gets mixed with the endosomal lipids, the $H_{II}$ phase becomes destabilized) acts as the driving force pushing the DNA into the cytosol.

In both pathways, after transfection of one dsDNA fragment, fusion can continue with another pore formation step to release the next fragment. At this point, either the lipoplex is already fully destabilized and the remaining cargo gets quickly transfected as well, or fusion halts in a prolonged hemifused state and some of the dsDNA remains trapped. In total 16 of the possible 20 dsDNA fragments (four fragments, five replicates) were successfully transfected in this setup. If complete fusion was achieved (*Figure 1V*), all dsDNA were at the opposing side of the bilayer. No leakage of dsDNA at the endosomal lumen was ever observed. Once transfected, although most of the dsDNA fragments still associated with the membrane (potentially stabilized by some of the cationic lipids from the lipoplex that come along), they were no longer enveloped by lipoplex components. The decomplexation of the dsDNA correlates well with the behaviour of lipoplex fusion in vitro by Rehman et al., where they describe that the DNA after transfection is no longer associated with lipoplex components (*ur Rehman et al., 2013*).

To quantify the overall transfection events further, we monitored the relative angle of the dsDNA fragments with respect to the membrane normal (*Figure 3—figure supplements 1–5*). In all cases, the first transfected fragment of dsDNA was either perpendicular to the bilayer normal, or close to the starting orientation (90° and 45° respectively). Remarkably, in some of the simulations the lipoplex reoriented itself such that the following transfection event(s) took place with the dsDNA in the parallel orientation, that is switching between the perpendicular and parallel pathway. This remarkable rearrangement might find its origin in the fact that the lipoplex core is suspended similar to a ball bearing, characterized by a low friction of the DNA/lipid core with respect to the coating layer. The removal of lipoplex material during fusion, which typically takes place on one side of the hemifusion diaphragm, might result in a torque causing the observed reorientation.

## DxTAP unsaturation is mandatory for efficient transfection

Having established two major fusion pathways leading to successful DNA transfection, next we consider the role of lipid composition in this process. Experimental studies reveal that lipid tail saturation and length, as well as the chemical composition of the headgroup of the lipoplex play an important role (*Koynova et al., 2009*; *Ma et al., 2007*; *Fletcher et al., 2006*). In this paragraph we aim to unravel the molecular influence of lipid tail saturation on lipoplex fusion and transfection efficiency. To this end, we constructed two additional versions of our lipoplex, replacing the mono-unsaturated DOTAP either by its fully saturated counterpart DPTAP (i.e., replacing the oleoyl tails by palmitoyl tails), or by a double unsaturated analogue DLiTAP (linoleoyl tails), whilst maintaining the same 1:4 DxTAP:DOPE ratio and target endosomal membrane composition (*Figure 2*, short lipids). The fusion experiments were set up in the same manner as before with five repeats per condition, each spanning ten microseconds.

For the DPTAP based lipoplex we never observed transfection, even though the stalk was maintained. The internal structure of the lipoplex was found to be somewhat unstable, with some of the channels fusing with each other and losing their classical $H_{II}$ hexagonal packing. Nevertheless, no DNA was lost from the complex. The orientation of the dsDNA was mainly parallel with respect to the bilayer, except for short periods of internal reconfiguration upon loss of the hexagonal unit cell

(*Figure 3—figure supplements 1–5*). The DLiTAP lipoplex, on the other hand, underwent fusion and showed cargo transfection in four out of five replicate simulations. The total amount of dsDNA fragments translocated amounted to 10 out of 20, somewhat less compared to the DOTAP lipoplex (*Figure 3*). Similar to DOTAP, both parallel and perpendicular fusion pathways were observed, as quantified by the dsDNA orientation analysis (*Figure 3—figure supplements 1–5*). Based on these results, it appears that unsaturated lipids complexing the dsDNA are required for successful fusion

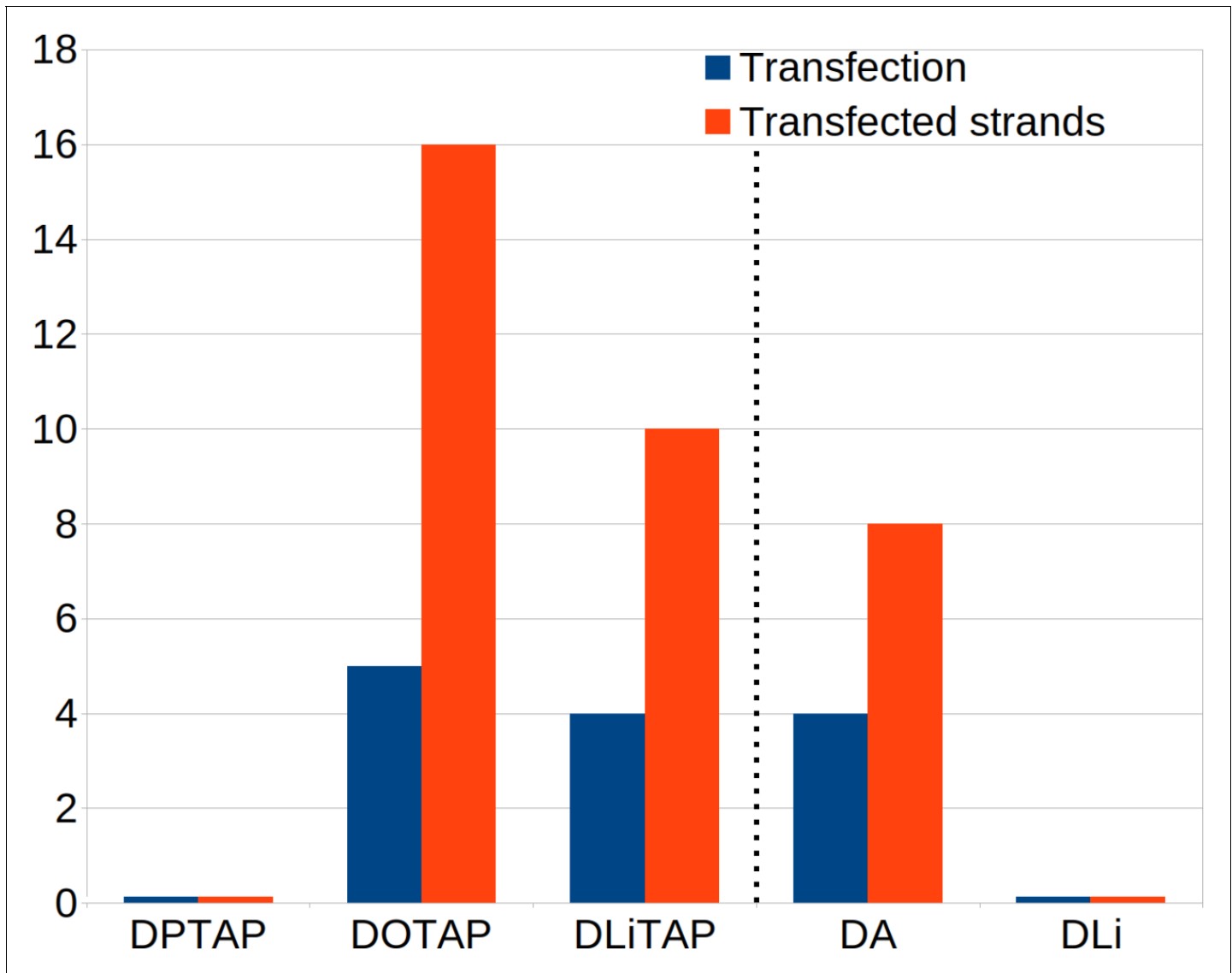

**Figure 3.** Effect of lipid composition on transfection efficiency. The number of successful transfections and amount of transfected DNA considering different lipid compositions. Each composition had 5 repeats with four dsDNA fragments each. Therefore the maximum number of transfections is five and the maximum amount of transfected fragments is 20 per condition. Left of the dotted line are the fusion results for varying lipoplex formulations on a short lipid membrane. For these formulations the lipoplex formulation was 4:1 DOPE/DxTAP. On the right of the dotted line the results for varying endosomal membrane compositions with constant lipoplex formulation (4:1 DOPE/DOTAP) are displayed. The alternative endosomal membrane models are enriched in poly-unsaturated DA or di-unsaturated DLi lipids (*Figure 2*).

The online version of this article includes the following figure supplement(s) for figure 3:

**Figure supplement 1.** Hydration/transfection, lipid mixing and orientation of the dsDNA DOPE:DPTAP + short lipid endosome bilayer.

**Figure supplement 2.** Hydration/transfection, lipid mixing and orientation of the DOPE:DOTAP + short lipid endosome bilayer.

**Figure supplement 3.** Hydration/transfection, lipid mixing and orientation of the DOPE:DLiTAP + short lipid endosome bilayer.

**Figure supplement 4.** Hydration/transfection, lipid mixing and orientation of the DOPE:DOTAP + DLi containing endosome bilayer.

**Figure supplement 5.** Hydration/transfection, lipid mixing and orientation of the DOPE:DOTAP + DA containing endosome bilayer.

and transfection. This agrees with fusion experiments between bilayers/vesicles, from which it has been suggested that more negatively curved lipids and or lipids with a decreased bending modulus favor formation of the fusion stalk and subsequent hemifusion diaphragm (*Markvoort and Marrink, 2011*; *Teague et al., 2002*; *Aeffner et al., 2012*; *Smirnova et al., 2010*; *Fan et al., 2016*). Since the bending modulus tends to decrease with the amount of unsaturations in the tails (*Levine et al., 2014*; *Pan et al., 2015*), one could naively anticipate an even further increased fusion efficiency for DLiTAP compared to DOTAP, contrary to our findings. In our understanding, this can be explained by the fact that unsaturation in the lipid tails does not only stabilize the stalk and other fusion intermediates, but also the $H_{II}$ phase itself (i.e. stabilize the lipoplex, preventing the release of cargo).

## Target membrane composition severely affects fusion rate

Our results point to a clear role for the composition of the lipoplex, raising the question whether the target membrane composition has a similar effect. To this end, we investigated the role of lipid composition of the endosomal membrane on the fusion efficiency of our lipoplex models. The precise lipid composition of the endosomal membrane is not known, and varies between cell types and level of maturation. Focusing on the role of lipid tails, we considered two additional endosomal membrane models, including some longer tailed lipids (diC16:0, dipalmitoyl, DP) and lipids with an increasing amount of unsaturation (diC18:2, dilinoleoyl, DLi and diC20:4, diarachidonyl, DA; *Figure 2*). We selected the most fusogenic lipoplex containing the DOTAP lipids, and performed again five repeats lasting ten microseconds each. Compared to the endosomal membrane model consisting of shorter lipids, described above, the fusion efficiency drops. In case of the endosomal membrane containing DLi lipids, no fusion events were observed at all, and the initial stalk state remained stable. For the endosomal membrane containing DA lipids, successful fusion and translocation of dsDNA was observed, but at a lower efficiency (*Figure 3*). When transfection occured, the mechanism was similar as described before as judged from the analysis of lipid mixing and dsDNA orientation (*Figure 3—figure supplements 1–5*). A possible explanation for the reduced fusion rates with the alternative endosomal membrane models is the increased stability of the bilayer by the longer lipid tails.

## Larger lipoplexes are more stable

After performing the fusion experiments with the small lipoplex (4 × 24 bp) and membrane, we expanded to a larger lipoplex (12 × 48 bp) and target membrane to investigate the effect of size on the fusion process (*Figure 4* and *Figure 4—figure supplement 1*). For the membrane conditions we selected the most reactive lipoplex formulation (DOTAP) and membrane composition (short lipids; *Figure 2*). The large solvated lipoplex was, like the smaller lipoplex, stable over a period of 10 µs and adopted an $H_{II}$ crystal lattice (*Figure 4A*). A difference compared to the smaller lipoplex, is that the large lipoplex contained a mixture of open and closed channels, with two of the twelve channels being continuous with the solvent on one end and one of them being open on both ends. The large lipoplex also contained additional channels connecting the $H_{II}$-channels, somewhat resembling a cubic phase. These connecting channels were large enough for ions, water and lipids to flow through, but the dsDNA remained strictly in the $H_{II}$ packing. We found that these connecting channels can have different origins. Some channels can occur when the equilibration of the $H_{II}$ phase is too short (50 ns); slightly elongating the preparation time removed such channels in the final periodic lipoplex (*Figure 4—figure supplement 1C*). However, increasing lipoplex hydration (lipids: waters$_{AA}$ 1:12) resulted in stabilization of the connecting channels (*Figure 4—figure supplement 1C*). In addition, connecting channels in the coated lipoplex appeared to be stable for at least 10 µs, and the coating process itself introduced new channels. Therefore our data suggest that these channels could be kinetically trapped structures, but also represent thermodynamic equilibrium structures depending on the exact state conditions. This is not unexpected if you consider the phase diagrams of lipid mixtures, which show a rich variety of inverted, cubic, and sponge phases that can interconvert as a function of temperature, hydration and other conditions.

Fusion was, as for the small complexes, initiated from a preformed stalk with five repeats each spanning ten microseconds. In contrast to the transfection behaviour on small scale, the larger lipoplex did not show any transfection events during this time period. We did observe a reorientation of the lipoplex from an initial parallel state to a state in which the dsDNA is oriented perpendicular to

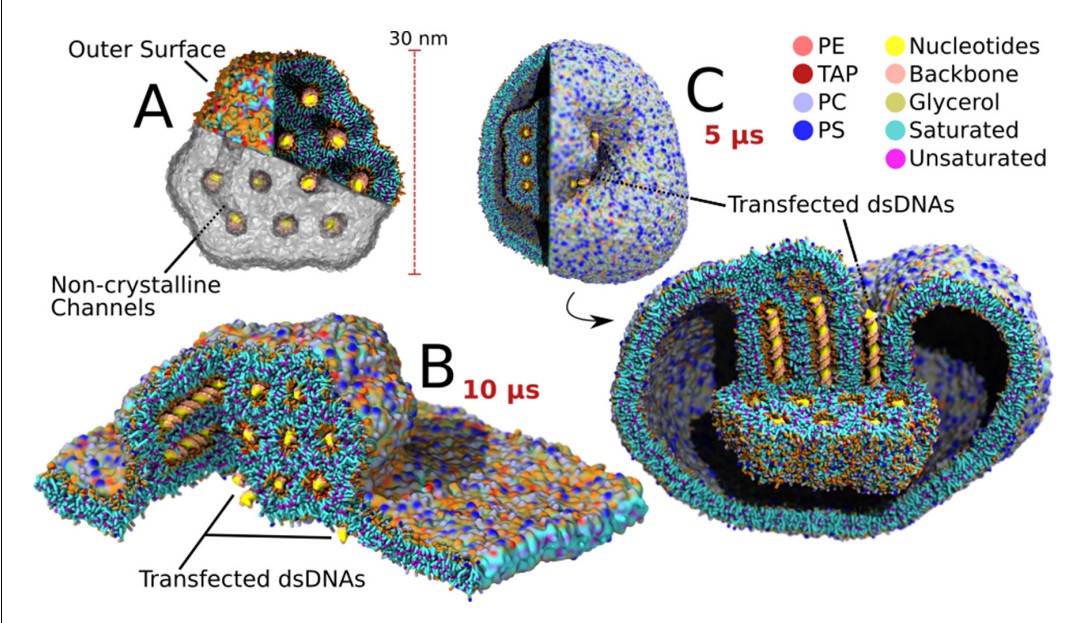

**Figure 4.** Transfection of large lipoplexes. The large lipoplex (A) showed the same stable H$_{II}$ structure as the small lipoplex with dsDNA inside the aqueous channels. Small additional connecting channels are also present, as indicated. Transfection was performed on top of a large endosomal model bilayer patch (B) and from within a model endosomal vesicle (C). In contrast to the small lipoplex, fusion did not spontaneously occur and had to be initiated using a biasing potential on one of the channels. After release of the biasing potential the dsDNA inside the pulled channel, and all dsDNA connected through connecting channels, transfected (3 DNA fragments transfected in both lamellar and vesicular system upon pulling of the same initial channel).

The online version of this article includes the following figure supplement(s) for figure 4:

**Figure supplement 1.** Large lipoplex channel structures with dsDNA inside.
**Figure supplement 2.** Reorientation of the large lipoplex in the endosome.

the membrane surface, whilst increasing the contact area (*Figure 4—figure supplement 2*). The lack of transfection for large lipoplexes raised the question whether this inhibition was of thermodynamic or kinetic nature. Therefore we performed five extra repeats of 10 µs, adding another biasing potential after formation of the stable hemifused complex to initiate the release of the first dsDNA fragment (see Materials and methods). The biasing potential was released after the first fusion event, that is opening of the channel. This resulted in transfection of the dsDNA inside the channel which was opened and all the dsDNA fragments in channels which shared a connection with it (*Figure 4B*). In total, 15 fragments were transfected (3 DNA fragments for each of the five repeats). In general, decomplexation of the dsDNA was much slower compared to the small lipoplex case, and mainly proceeded by the unzipping pathway (*Figure 1IVa*). Thus it appears that cargo delivery can also take place with larger lipoplexes, but that kinetic trapping is more likely to occur in either a surface contact or hemifused state.

To explain the difference in reactivity of the large lipoplex with respect to the smaller one, it seems plausible that the curvature of the lipoplex plays an important role in fusion efficiency. The same is observed in case of fusion between small liposomes and membranes (*Kawamoto et al., 2015*). However, there is a second issue that should be considered, namely the overall stability of the lipoplex. This stability is affected by the geometric discrepancy between the optimal configuration for the lipoplex coating and the inner core. Whereas the optimal configuration of the coating would be a sphere, satisfying minimal curvature constraints, the optimal configuration of the core is an H$_{II}$ phase which does not have uniform curvature on its circumference. From these two requirements it follows that there will be an interfacial tension between the coating and the core, in addition to the interfacial tension of the coating with the surrounding solvent. If we compare the geometry of the small lipoplex to that of the large one there is a clear difference in the outer angles of the core. Considering a 2D projection along the channel axis, the small lipoplex has two 60° and

two 120˚ angles. The large lipoplex has six 120˚ angles. The bimodal angle distribution of the small lipoplex is further away from the preferred constant curvature of the coating than the larger one, resulting in a higher interfacial tension per lipid. Together, the lipoplex curvature stress and the interfacial tension between the core and coating layer explain the observed trend in reduced fusion efficiency upon increase in lipoplex size.

## Escape from the endosome

Finally, to study lipoplex fusion with endosomal membranes in a more realistic setting, we embedded a large lipoplex inside a small vesicle representing the endosome (*Figure 4C*). In principle, the curvature present in the endosomal membrane could further modulate the fusion pathway and efficiency. The lipid composition of the vesicular endosome consisted again of the short lipid variant (*Figure 2*). The vesicle measured 50 nm in diameter and the total system size exceeded 3 million CG beads (representing roughly 30 million atoms). To allow simulation of this large system size, the duration of the production run was reduced to 5 µs and only a single simulation was performed. As for the large bilayer system, no spontaneous fusion occurred, and a dsDNA fragment was pulled for initiation of transfection. Once the channel opened, dsDNA was transfected, followed by another two fragments that shared the same aqueous space via the connecting channels (*Figure 4C*). Opposed to the zipper-like fusion observed in the large bilayer experiment, this time fusion occurred in a mixed manner, combining the vertical alignment with the unzipping between neighbouring channels, resulting in a slow release (*Videos 2* and *3*). Although we can not extract generic behavior from this single experiment, the observed pathway points to a possible role of curvature on the preferred orientation of the lipoplex with respect to the endosomal membrane, with possible consequences for transfection efficiency.

## Conclusion

We were able to successfully simulate the lipoplex mediated transfection of dsDNA over a model endosomal membrane and observed two profoundly different fusion pathways. Release of the dsDNA by rapid unzipping of the $H_{II}$ phase occurs when the dsDNA lies perpendicular to the membrane normal, whilst an ejection-like or slow unzipping release of dsDNA is observed when the dsDNA is oriented parallel to the bilayer normal. Transitions between these states are also possible along the entire fusion pathway. Interestingly, our results for the small lipoplex indicate that transfection of the first dsDNA fragment triggers the release of a substantial amount, if not all, of the

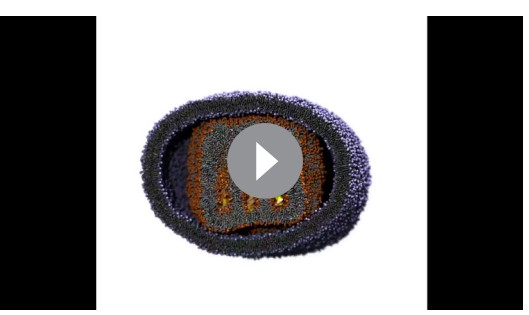

**Video 2.** Large lipoplex-vesicle transfection. Fusion of the large lipoplex from inside the endosomal vesicle. The headgroups of the endosomal and lipoplex lipids are subsequently blue and orange. The endosomal lipids have dark grey tails and the lipoplex lipids have light grey tails. Yellow and pink are used for subsequently the bases and backbone of the dsDNA. No bias was applied during this simulation, though opening of the first channel occurred by a biasing potential as described in the methods section.
https://elifesciences.org/articles/52012#video2

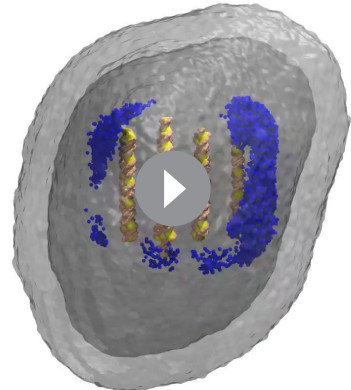

**Video 3.** Reorientation of lipoplex in vesicle. Components of the lipoplex closer than 4 nm to the endosome are colored blue, the endosomal lipids are grey and the dsDNA inside the lipoplex is yellow/pink. Within 1 µs the lipoplex reorients itself such that its $H_{II}$-channels are roughly aligned with the membrane normal in the contact region.
https://elifesciences.org/articles/52012#video3

remaining genetic material likely caused by further destabilization of the lipoplex once cargo gets released.

Considering that the lipid composition of the endosomal membrane is not up for control, the differences between the lipoplex formulations appear to be most interesting from a rational design point of view. However, the endosomal bilayer composition varies per species and cell type, therefore understanding lipoplex-endosome interactions are of equal importance. We found that the transfection efficiency is sensitive to minor changes in both lipoplex and endosomal membrane composition, which is in line with experiments (*Huang and Li, 1997*). The highest amount of transfection was observed for the DOTAP:DOPE lipoplex. No transfection was observed for the DPTAP:DOPE complex, and DLiTAP:DOPE showed intermediate fusogenicity. Interestingly, the same lipid type modifications (e.g., presence of poly-unsaturated tails) can have both stabilizing and destabilizing effects on fusion efficiency, making rational predictions for optimal delivery vehicles difficult. Moreover, our results point to an important role of lipoplex size, with the larger lipoplex being less fusogenic. This we attribute to an increase of lipoplex stability causing slower lipid mixing of the lipoplex and the target membrane. In addition, the size and shape of the lipoplex affect the way the complex orients itself and interacts with the (curved) endosomal membrane. In this study we focused our efforts on exploring the effect of varying lipid tails, keeping many other parameters of the lipoplex constant. Future endeavors could investigate the role of lipid headgroups and charge ratios, which are known to affect transfection (*Eastman et al., 1997*; *Ma et al., 2007*). In addition, the DNA/lipid ratio and size of the DNA fragments could play a role. It would be interesting to see to what extent changing such parameters alters the observed fusion pathways and kinetics. To provide further insights into the process, the observed mechanisms need to be cast into a continuum type of description (*May and Ben-Shaul, 2004*; *Hamm and Kozlov, 2000*) from which the competing energetic driving forces can be more straightforwardly extracted.

Eventually, our in-silico predictions need experimental validation, although resolving fusion pathways at the molecular level remains highly challenging. Time resolved SAXS in combination with single-molecule fluorescence microscopy could be used to probe the relative orientation of the lipoplex and embedded genetic material during the transfection process. Furthermore, leakiness assays could be used to test whether or not endosomal material leaks out concomitantly, for example via transient pore formation. Although the lipoplex fusion process described here does not involve such transient pores, we note that the presence of open channels in the lipoplex (*Figure 4—figure supplement 1*) implies that endosomal material can become transfected along with the dsDNA even if the fusion itself is non-leaky. Besides, leaky fusion pathways have been observed in previous simulation studies between lipid membranes, and appear highly dependent on both lipid compositions and local stress conditions (*Markvoort and Marrink, 2011*). Therefore, it seems plausible that such transient pores could also form during lipoplex fusion.

To conclude, we demonstrated that detailed computer simulations of the fusion between a lipoplex and model endosomal membranes is nowadays possible, opening the way for systematic studies using the more advanced lipoplex formulations currently available (*Barrán-Berdón et al., 2014*; *Al-Dulaymi et al., 2019*; *Severino et al., 2015*; *Leite Nascimento et al., 2016*). Besides, the observed fusion pathways could be of generic importance for uptake of lipid-complexed nanoparticles into cells following endocytosis.

## Materials and methods

### Building the small periodic lipoplex

We followed a three step procedure for setting up the lipoplex-membrane fusion experiment. First we constructed the all-atom (AA) structure of the 24 base pairs dsDNA sequence ([CGCGAA TTCGCG]$_2$) using the B-DNA sequence to structure builder at the website of IIT Dehli (http://www.scfbio-iitd.res.in/software/drugdesign/bdna.jsp; *Arnott et al., 1976*). We utilized Martinize to coarse-grain (CG) the AA structure to a Martini stiff-dsDNA force field and structure (*Figure 1—figure supplement 1A*; *Uusitalo et al., 2015*). To build the periodic inverted hexagonal phase we roughly used the procedure described by *Corsi et al. (2010)*. The CG DNA was placed on top of a bilayer constructed with the bilayer builder *insane* which contained DOTAP and DOPE at a 1:4 ratio (*Figure 1—figure supplements 1B1*; *Figure 2*; *Supplementary files 1*, *2*; *Marrink et al., 2007*;

*Wassenaar et al., 2015*). This system was duplicated in its *x* dimension, yielding a single membrane with two dsDNA fragments stacked on top of it with a spacing of 6 nm with respect to each other, using GROMACS gmx editconf (*Abraham et al., 2015*). The system was hydrated to achieve a hydration of two atomistic waters per lipid and a salinity of 150 mM NaCl. In addition, each system was charge neutralized by exchanging random water beads for the appropriate counterion ($Na^+$/ $Cl^-$). To prevent freezing of the water we added 10% anti-freeze particles (*Marrink et al., 2007*). The bilayer system was stacked on top of itself in the *z* dimension to obtain system of two bilayers and four dsDNA fragments (*Figure 1—figure supplement 1C*). The stacked system was energy minimized and equilibrated using GROMACS 5.1.5. For energy minimization we used the steepest descent algorithm and for the equilibration we used the default Martini settings making use of a two fs time step up to the point that numerical stability was achieved (*de Jong et al., 2016*). The Verlet cut-off scheme was used with a 1.1 cut-off for both the coulombic (reaction-field) and van der Waals interactions. We used v-rescale for the thermostat at 310 K, coupling the DNA, lipids and solvent in separate groups. Pressure coupling was performed using the Berendsen barostat (*Berendsen et al., 1984*) for anisotropic systems. The production run made use of a 10 fs time step and the pressure coupling was switched to Parrinello-Rahman (*Parrinello and Rahman, 1981*). Within 50 ns the system changed from a stacked bilayer assembly into a stable inverted hexagonal phase ($H_{II}$). This general procedure was used for all lipoplex formulations. However, in the presence of a stable lamellar bilayer the pressure coupling was set to semi-isotropic, and in the case of a free particle in solution the pressure coupling was handled isotropically. A detailed description including a step by step tutorial is described in our own chapter in *Bonomi and Camilloni (2019)*.

## Building the small solvated lipoplex

Once a stable periodic $H_{II}$ phase was obtained, we put the complex in a bigger box to add another layer of lipids around it. The surface of the lipoplex was approximated to be a parallelepiped to simplify geometric calculations. We used a symmetrical bilayer system to calculate the area per lipid (APL) of the DOPE, DOTAP mixture (*Supplementary file 2*). The approximated area of the lipoplex was divided by the APL of the lipid mixture. The calculated amount of lipids were added in a hollow cube around the naked lipoplex using PACKMOL (*Martínez et al., 2009*). Water and ions were added as before. To test for the accuracy of this method, we also tried half and double the calculated amount and investigated lipoplex stability (*Figure 1—figure supplement 3*). The system was solvated, energy minimized and equilibrated using the same settings as before except for the pressure coupling which was set to isotropic. The production run spanned 10 µs. The radius of gyration (gmx gyration) and hydration (in-house VMD based selection counter) of the lipoplex were analyzed over time, to evaluate the equilibrated state (*Figure 1—figure supplement 2*). This general procedure was used for all lipoplex formulations. The DPTAP and DLiTAP containing lipoplexes were generated from the DOTAP lipoplex by exchanging the DOTAP lipid force field with either DPTAP or DLiTAP plus an extra step of equilibration (5 µs). The small lipoplex had a diameter of ~20 nm and the box dimensions were 28.1 28.1 19.8 nm for xyz respectively resulting in a total of 118,718 beads. The periodic boundaries were set to dodecahedron. A detailed description including a step by step tutorial is described in our own chapter in *Bonomi and Camilloni (2019)*.

## Building the large lipoplex

The same procedure to build the small lipoplexes was used to construct the large lipoplex. However, instead of four dsDNA fragments these lipoplexes contained twelve dsDNAs which had a length of 48 base pairs ([CGCGAATTCGCG]$_4$) each. The final lipoplex had a diameter of ~30 nm.

## Building the lipoplex-bilayer system (small)

To investigate the fusion behaviour of lipoplexes with membranes, we built several lipoplexes with varying composition. We prepared a system which has long polyunsaturated tails (DAPC), medium unsaturated tails (DLiPC) and short single unsaturated tails (DYPC) in the bilayer as its unsaturated component (*Figure 2*). In the case of the short lipid bilayer, all the lipids were shorter. The solvated lipoplex was placed on top of the appropriate bilayer using insane (*Wassenaar et al., 2015*). The systems were solvated and salinized, energy minimized and equilibrated as before, but the pressure coupling was set to semi-isotropic in the *xy* and *z* dimensions (with the membrane normal in *z*). After

the system was equilibrated and the lipoplex adhered to the membrane, a pulling potential was added between ~10 lipids of the outer leaflet of the lipoplex in the proximity of the bilayer and vice versa. The initial center of geometry distance between the pulling groups was less than 3 nm and the pulling lasted for less than five ns (harmonic biasing force; constant velocity −1 m/s). After the initial stalk was formed, the biasing potential was removed. The system was equilibrated and a 10 µs transfection experiment was performed five times with random starting velocities for each membrane and lipoplex formulation. Again the pressure coupling was set to semi-isotropic. The small lipoplex bilayer box dimensions were 27.5 27.2 31.6 nm in xyz respectively with a total of 193,083 beads. A detailed description including a step by step tutorial is described in our own chapter in *Bonomi and Camilloni (2019)*.

### Building the lipoplex-bilayer system (large)

Using the optimal performing bilayer and lipoplex from the small lipoplex-bilayer experiments, we set up a system with a large lipoplex (DOPE:DOTAP; 12*48 bp) and the short lipids bilayer (*Figure 2*). Adhesion and stalk formation was performed identical to the small version of the experiment. However, after stalk formation two flavors of the experiment were conducted. In the first variant the experiment was left untouched after stalk formation for 10 µs. In the second variant of the experiment, we simulated until a stable fusion intermediate (no transfection) was obtained and then we initiated another pull. However, this time not pulling the tails of lipids in opposing leaflets together, but the heads of the lipids directly above and under the stalk/diaphragm. Thus pulling headgroups of lipids from inside one of the channels in the lipoplex, towards the headgroups of the opposing side of the bilayer. The initial centre of geometry distance between the two pulling groups was ~5 nm and lasted up to five ns (harmonic biasing force; constant velocity −1 m/s). This initiated a fusion pore from which transfection followed. After instantiation of the fusion pore the biasing potential was removed, allowing for unbiased transfection of the following fusion and transfection events. After the biasing potential was released five simulations were started from this point with random initial velocities, each running for 10 µs. The large lipoplex-bilayer box dimensions were 45.8 45.8 44.5 nm in xyz respectively, with a total of 749,538 beads.

### Building the lipolex-vesicle system (large)

As for the large lipolex-bilayer system, the best performing lipoplex and bilayer composition in the small experiments were used. The initial state of the vesicle was constructed using an in-house Python three script based on a spherical fibonacci spiral (available on request). The amount of lipids placed on the inside was corrected for the area difference between the inner and outer leaflet, and the APL of DLPC was used to calculate the amount of lipids needed. After placement of the DLPC, the system was solvated and salted using *insane* and the appropriate lipid force fields we added in the topology file. The same protocol used in *Qi et al. (2015)*, and *Risselada et al. (2008)* to equilibrate a vesicle was used, relaxing the leaflet asymmetry through pores. After equilibration of the inter-leaflet tension (100 ns; production-md-settings with isotropic pressure coupling) the vesicle was dehydrated and desalted. The lipoplex plus its surrounding ions, waters and DNA were placed at the center of the porated vesicle and the system was rehydrated and salted using *insane*. This system was equilibrated for 800 ns, allowing for the ion content to equilibrate as well, tension release due to leaflet asymmetry occured in a few ns. The biasing potential for the pore was removed, which rapidly caused closing of the pores (<10 ns). The whole complex was then simulated for 1 µs during which the lipoplex adhered to the inside of the vesicle. From here the stalk was initiated as before for the small and large lipolex bilayer simulations. As for the large lipoplex-bilayer simulations, the experiment was split in two, one being the unbiased simulation of the stalk and possible transfection (1 µs). The second experiment included another biasing potential to initiate channel-vesicle fusion. The pulling procedures used were exactly the same as for the large lipolex-bilayer systems and after release the unbiased lipoplex-vesicle simulation spanned 5 µs. The large lipoplex-vesicle box dimensions were 72.8 72.8 72.8 nm in xyz respectively, with a total of 2,994,561 beads.

### Analysis

The dsDNA spacing in the small solvated lipoplex was calculated using the center of mass (COM) distance between the first neighbours of each dsDNA over the last 2 µs of the solvated lipoplex

(n = 5). We made a Python three script to analyze the angle of the dsDNA with respect to the z unit vector (normal to the average bilayer plane), using the COM of the first and last base pairs to construct the dsDNA orientational vector. The script treats the dsDNA termini as equals, therefore no higher angle with the membrane normal than 90° can be obtained. The same VMD based counting script used for the calculation of hydration was used to calculate the lipid contacts and dsDNA-water contacts (cutoff 1.2 nm). The dsDNA-water contacts were used for the automated transfection detection script, which reported a transfection event whenever dsDNA hydration rapidly increased (more than 200 CG waters in less than 400 ns). The automated transfection detection was manually checked on five simulations, both with and without fusion, and was always correct. We used all the information above to investigate the angle with respect to transfection, as well as the importance of lipid mixing. VMD 1.9.3 was used to render the images and videos (*Humphrey et al., 1996*). The Tcl script used in combination with VMD to perform fast selection counting, and the Python script to perform the angle analysis and fusion detection are available in the repository. In general Python three and MDAnalysis were used extensively in the analysis and visualization (*Gowers et al., 2016*; *Michaud-Agrawal et al., 2011*).

## Acknowledgements

We would like to thank Prof. dr. D Hoekstra for his ideas and comments on the construction of the lipoplex model. His experience within the lipoplex field helped us to mature many of the design choices made in this research. We would also like to thank Dr. T Wassenaar and Dr. J Barnoud for helping with some of the problems we had during coding. SJM acknowledges funding from the ERC through an Advanced grant 'COMP-MICR-CROW-MEM'. This work was carried out on the Dutch national e-infrastructure with the support of SURF-SARA, and the Peregrine HPC at the University of Groningen.

## Additional information

### Funding

| Funder | Grant reference number | Author |
| --- | --- | --- |
| H2020 European Research Council | COMP-MICR-CROW-MEM | Siewert-Jan Marrink |

The funders had no role in study design, data collection and interpretation, or the decision to submit the work for publication.

### Author contributions

Bart MH Bruininks, Conceptualization, Data curation, Software, Formal analysis, Investigation, Visualization, Methodology; Paulo CT Souza, Conceptualization, Supervision, Validation, Methodology; Helgi Ingolfsson, Conceptualization, Methodology; Siewert J Marrink, Resources, Supervision, Funding acquisition, Methodology

### Author ORCIDs

Bart MH Bruininks (iD) https://orcid.org/0000-0001-5136-0864
Siewert J Marrink (iD) https://orcid.org/0000-0001-8423-5277

### Decision letter and Author response

Decision letter https://doi.org/10.7554/eLife.52012.sa1
Author response https://doi.org/10.7554/eLife.52012.sa2

## Additional files

### Supplementary files

• Supplementary file 1. DOTAP_validation.

- Supplementary file 2. lipoplex_lipid-properties.
- Transparent reporting form

## Data availability

All raw data of fusion experiments and analysis software amount to TBs of data, so are available upon request. A data package has been prepared and deposited to Dryad, under the DOI https://doi.org/10.5061/dryad.fqz612jq4.

The following dataset was generated:

| Author(s) | Year | Dataset title | Dataset URL | Database and Identifier |
|---|---|---|---|---|
| Bruininks BMH, Souza PCT, Ingólfsson H, Marrink SJ | 2020 | A molecular view on the escape of lipoplexed DNA from the endosome (no trajectories) | https://doi.org/10.5061/dryad.fqz612jq4 | Dryad Digital Repository, 10.5061/dryad.fqz612jq4 |

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
