## [Decision Letter]

**Acceptance summary:**

The work by Marrink's group proposes a molecular mechanism of the endosomal release of genetic material from a non-viral vector. This mechanism is novel and will help design non-viral vectors stimulating gene therapy research. The work, based on the state-of-the-art large-scale molecular dynamics simulations, captures the entire fusion pathway of a non- viral vector with the endosomal membrane up to the point where the genetic material escapes from the endosome. Systematic variation of the lipid composition of both the non-viral vector and the endosomal membrane reveals the molecular factors that play a key role during gene transfection. The work is not only of interest to researchers working in the field of gene delivery but also to those working on nanoparticle uptake in a broader sense including drug delivery. Furthermore, the work is also highly innovative from the modeling perspective, which includes building and validation of the non-viral vectors as well as the large system complexity.

**Decision letter after peer review:**

Thank you for submitting your article "A molecular perspective on lipoplex mediated genetic outbreak from the endosome" for consideration by *eLife*. Your article has been reviewed by two peer reviewers, and the evaluation has been overseen by a Reviewing Editor and Olga Boudker as the Senior Editor. The following individuals involved in review of your submission have agreed to reveal their identity: Nejat Duzgunes (Reviewer #2); Sylvio May (Reviewer #3).

The reviewers have discussed the reviews with one another and the Reviewing Editor has drafted this decision to help you prepare a revised submission.

Summary:

The work represents an outstanding effort to use Molecular Dynamics to get insight into the mechanisms of DNA delivery to cells. The authors perform coarse-grained MD simulations to study the interaction between a lipoplex and an extended lipid bilayer or a small enveloping vesicle. In computational experiments using a pre-formed stalk structure as the starting state, fusion and subsequent release of short dsDNA segments are observed for certain combinations of lipids in the lipoplex and the interacting membrane. Release events are interpreted in terms of achieving transfection following the endosomal escape of DNA.

Having an overall positive opinion on the study, the reviewers formulated several concerns about the presentation of the results. The three major requests, which, according to the reviewer comments and my own impression have to be addressed are: (i) the physical mechanisms underlying the observed and described events have to be discussed, (ii) a discussion of predictions, which can be made, based on the computational results, on the behavior of real membranes differing from the modelled ones in composition and physical properties (ii) the title and the introducing sections of the article somewhat overemphasize the generality of the content, which should be downplayed.

Essential revisions:

Most specifically, the reviewers requested to addresses the following issues:

1) The study represents a description of simulation results, which raises questions about the physical mechanisms behind the observed events. Since this is not an experimental but rather a computational study based on modelling the inter- and intra- molecular physical interactions, the physical forces behind the observations should be explicitly discussed. Particularly, the end of paragraph two in subsection "Two distinct mechanisms leading to gene transfection" states that DNA fragments are ejected like the unloading of a syringe. Why are they ejected? What causes the DNA to move away from the cationic lipids in the tube? Another example, larger lipoplexes appear to have small additional connecting channels (Figure 3). Why do they appear? Do they represent thermal equilibrium with defects or are these kinetically trapped nonequilibrium structures due to the lipoplex preparation protocol? Do they disappear if the final wrapping layer is supplemented by a few lipids? Yet another example, why is the lipoplex reorienting itself to ensure parallel or perpendicular orientation of the DNA? What is the origin of the torque that acts of the lipoplex?

The computational results demonstrate that initiating DNA release takes longer for larger lipoplexes. What is the physical reason for this effect? Could it be that the fusion rate is set by the curvature of the lipoplex lipid shell, which is quite high even the large 30-nm lipoplex analyzed in this study? If this is the case, could it be that for realistically large lipoplexes fusion events must be expected to take much longer or not happen at all?

2) In the computational model, the structure of lipoplexes is characterized by three parameters: the amounts of the cationic lipids, the helper lipids, and DNA. The authors calculations have been performed for two particular sets of these values (small and large lipoplexes, both charge compensating), but there are many other choices that may be relevant for real systems (if, for example, the lipoplex is prepared in the presence of excess DNA or excess cationic lipid). The resulting lipoplex may become positively or negatively overcharged in these cases, with possible implications on the fusion pathways. In addition, slightly adjusting the number of lipids at fixed DNA may create frustration and/or tension of the outer wrapping layer, this may also modify the pathway and kinetics of fusion. The discussion of these issues will be beneficial for the impact of the work.

3) Presenting the results of the study in the title as "perspective on gene therapy" sounds somewhat too ambitious since the work addresses only one, albeit important, aspect of the pathway toward transfection for one specific type of non-viral vector. In addition, the relation of the expression "Genetic outbreak from the endosome" to the content remains unclear.

Along the same lines, the sentences in the Abstract and throughout the paper concerning the proposed mechanism should be reformulated in a more suggestive rather than confirmative way since the simulated system, obviously, differs in their composition and other details from real intracellular membranes.

4) Finally, several introductory statements need to the formulated in a more accurate way:

– Lipoplexes can trigger a non-specific immune response; so, this sentence should be changed and a reference should be given.

– Another major downside of lipoplexes is their potential toxicity, especially if they are highly positively charged.

– DOTAP is not 1,2-dipalmitic...

– The van Meer et al. paper shows mostly PC, PE and SM. Moreover, their Figure 3 shows that the lumen-exposed lipids are primarily PC, SM and glycosphingolipids. How will this change the conclusions of the paper?

– The authors should discuss how their observations relate to the transfection of a much larger DNA molecule.

---

## [Author Response]

Essential revisions:Most specifically, the reviewers requested to addresses the following issues:1) The study represents a description of simulation results, which raises questions about the physical mechanisms behind the observed events. Since this is not an experimental but rather a computational study based on modelling the inter- and intra- molecular physical interactions, the physical forces behind the observations should be explicitly discussed.

We agree that the underlying forces of the physical mechanisms would be valuable to discuss. However, we would like to stress that this is far from trivial, despite the fact that we have access to all inter and intramolecular interactions. In the end, the physical mechanisms or driving forces are the result of all these complex interactions. Ideally, one would extract the potentials of mean force (PMF) along some chosen reaction coordinate. However, this is complicated as the fusion of the lipoplex and subsequent release of DNA are both highly non-linear events for which a simple reaction coordinate is not easily defined. Defining suitable reaction coordinates (and subsequent PMF computation) between simple planar membranes or liposomes has already been proven to be a big challenge in the field of computer simulations (see for instance Markvoort and Marrink, but I know the reviewers are well aware of such issues). Even if a meaningful reaction coordinate could be found, to compute a converged PMF for a system of this size and complexity is out of reach even for a CG model like Martini.

Alternatively, one could simply look at the temporal evolution of the interaction energies (e.g., lipid or DNA solvation, lipid-lipid interaction) to extract certain trends. We did, but found that this can be easily misleading due to the fact that many processes occur at the same time. What, in our view is needed but outside the scope of our work, is to cast the observed mechanisms into a continuum type of description (our hope is that the reviewers might jump on this, given their expertise in this area!) from which the competing energetic driving forces can be more straightforwardly extracted. In the revised manuscript, we did try to at least supply some additional reasoning about potential driving forces.

Particularly, the end of paragraph two in subsection "Two distinct mechanisms leading to gene transfection" states that DNA fragments are ejected like the unloading of a syringe. Why are they ejected? What causes the DNA to move away from the cationic lipids in the tube?

Visual inspection of the unloading process (shown in the Video 1) reveals that the dsDNA is moving out of the channel at the same time of the merging of the lipids that line the channel with the endosomal membrane. This suggests that the release of curvature stress of the lipoplex (through fusing with the endosomal membrane) is the driving force, and that the DNA is simply pushed away during this process resulting in its ejection. Note that the dsDNA, after being transfected, does have a tendency to stay adsorbed to the endosomal membrane, likely stabilized by some of the cationic lipids that come along. Eventually, all dsDNAs are expected to desorb and escape into the cytosol (as the cationic lipids will dilute across the endosomal membrane, as well as just for entropic reasons). We added our reasoning on the release of the dsDNA to the text.

Another example, larger lipoplexes appear to have small additional connecting channels (Figure 3). Why do they appear? Do they represent thermal equilibrium with defects or are these kinetically trapped nonequilibrium structures due to the lipoplex preparation protocol? Do they disappear if the final wrapping layer is supplemented by a few lipids?

To be able to answer this question we performed additional simulations of the large periodic lipoplex to obtain a better understanding of the origin and behaviour of such channels. We found that, by elongating the equilibration simulation for the formation of the inverted hexagonal phase, we could completely remove the connective channels under the original conditions, suggesting they are kinetically trapped non-equilibrium structures. However, the channels could be stabilized by adding only slightly more water (15% extra), suggesting that these channels could also represent thermodynamic equilibrium structures depending on the exact state conditions. This is not unexpected if you consider the phase diagrams of lipid mixtures, which show a rich variety of inverted, cubic, and sponge phases that can interconvert as a function of temperature, hydration and other conditions.

Furthermore, we took a closer look at the internal channel dynamics in the simulations of the solvated lipoplex. We found that indeed during the coating process new channels were formed which were not directly inherited from the periodic stage. These channels were very stable on the accessed timescale of 10 Âµs. Thus it appears that the coating process itself can also affect the internal stability of the DNA/lipid phase. Given the long time it takes to setup and equilibrate the lipoplexes, we did not investigate the role of the coating layer in more detail, but this is surely a factor that could play a role in the overall transfection efficiency. We think this is also worth further exploration in wet-lab experiments. The new insights on the stability of the connecting channels are discussed in the revised manuscript.

Yet another example, why is the lipoplex reorienting itself to ensure parallel or perpendicular orientation of the DNA? What is the origin of the torque that acts of the lipoplex?

Sadly enough the problem with answering this question is the same as for finding the exact reason for the dsDNA ejection. Our current sampling and lack of understanding of the phenomena does not allow us to give an experiment based explanation. However, we think it might find its origin in the fact that the lipoplex core is suspended in a fashion very similar to ball bearings. This could make the rotational friction of the DNA/lipid core with respect to the coating layer extremely low. The removal of lipoplex material during fusion, which typically takes place on one side of the hemifusion diaphragm, might be enough to result in a torque causing the observed reorientation. We discuss this in the revised manuscript.

The computational results demonstrate that initiating DNA release takes longer for larger lipoplexes. What is the physical reason for this effect? Could it be that the fusion rate is set by the curvature of the lipoplex lipid shell, which is quite high even the large 30-nm lipoplex analyzed in this study? If this is the case, could it be that for realistically large lipoplexes fusion events must be expected to take much longer or not happen at all?

Indeed it seems plausible that the curvature of the lipoplex shell plays an important role in fusion efficiency. The same is observed in case of fusion between small liposomes and membranes (Kawamoto et al., 2015). However, there is a second issue that should be considered, which is the overall stability of the lipoplex. This stability is affected by the geometric discrepancy between the optimal configuration for the lipoplex coating and the inner core. Whereas the optimal configuration of the coating would be a sphere, satisfying minimal curvature constraints, the optimal configuration of the core is an H_II_ phase which does not have uniform curvature on its circumference. From these two requirements it follows that there will be an interfacial tension between the coating and the core, in addition to the interfacial tension of the coating with the surrounding solvent. If we compare the geometry of the small lipoplex to that of the large one there is a clear difference in the outer angles of the core. Considering a 2D projection along the channel axis, the small lipoplex has two 60Ëš and two 120 ÌŠ angles. The large lipoplex has six 120Ëš angles. The bimodal angle distribution of the small lipoplex is further away from the preferred constant curvature of the coating than the larger one, resulting in a higher interfacial tension per lipid. Considering even bigger lipoplexes, the amount of curvature mismatch per lipid would decrease even more resulting in a lower reactivity per lipid/dsDNA. Together, the lipoplex curvature stress and the interfacial tension between the core and coating layer explain the observed trend in reduced fusion efficiency upon increase in lipoplex size. We added an extra paragraph to explain lipoplex stability with respect to size, based on our current understanding.

2) In the computational model, the structure of lipoplexes is characterized by three parameters: the amounts of the cationic lipids, the helper lipids, and DNA. The authors calculations have been performed for two particular sets of these values (small and large lipoplexes, both charge compensating), but there are many other choices that may be relevant for real systems (if, for example, the lipoplex is prepared in the presence of excess DNA or excess cationic lipid). The resulting lipoplex may become positively or negatively overcharged in these cases, with possible implications on the fusion pathways. In addition, slightly adjusting the number of lipids at fixed DNA may create frustration and/or tension of the outer wrapping layer, this may also modify the pathway and kinetics of fusion. The discussion of these issues will be beneficial for the impact of the work.

We fully agree that many additional factors come into play, and added a short discussion on this topic in the conclusion, pointing out that we focused on the nature of the lipid tails, but future investigation of the effects of charge compensation and different lipid/DNA ratio could be extremely beneficial to our understanding of transfection of these complexes.

3) Presenting the results of the study in the title as "perspective on gene therapy" sounds somewhat too ambitious since the work addresses only one, albeit important, aspect of the pathway toward transfection for one specific type of non-viral vector. In addition, the relation of the expression "Genetic outbreak from the endosome" to the content remains unclear.Along the same lines, the sentences in the Abstract and throughout the paper concerning the proposed mechanism should be reformulated in a more suggestive rather than confirmative way since the simulated system, obviously, differs in their composition and other details from real intracellular membranes.

We agree, and made changes throughout the text to solve this issue. We hope the article no longer reads to be more general than it is and tried to make it clear that all are conclusions based on the models that we made. We also focused more on the transfection and less on the gene therapy. This is also reflected in the revised title, which now reads: Escape of lipoplexed DNA from the endosome: a molecular perspective on transfection.

4) Finally, several introductory statements need to the formulated in a more accurate way:- Lipoplexes can trigger a non-specific immune response; so, this sentence should be changed and a reference should be given.- Another major downside of lipoplexes is their potential toxicity, especially if they are highly positively charged.

We added a reference and point out the issue of toxicity and non-specific immune responses in the Introduction.

- DOTAP is not 1,2-dipalmitic...

Thanks, changed to dioleoyl.

- The van Meer et al. paper shows mostly PC, PE and SM. Moreover, their Figure 3 shows that the lumen-exposed lipids are primarily PC, SM and glycosphingolipids. How will this change the conclusions of the paper?

Given the large complexity of the simulated systems, we decided to keep the endosomal membrane composition as simple as possible, with PC as main phospholipid. PS was added as it was hypothesized before that anionic lipids could play an important role in the transfection mechanism (Tarahovsky, Koynova and MacDonald, 2004). Clearly, real endosomal membranes are much more complex concerning lipid composition, and, to a large extent, the exact composition is unknown (especially concerning asymmetry) and varying between endosomal stage and cell type. To what extent these lipids could affect the potential fusion pathways is difficult to say, and needs to be addressed in follow up studies. We emphasized this aspect in the revised manuscript.

- The authors should discuss how their observations relate to the transfection of a much larger DNA molecule.

We touch the topic slightly when we discuss the connective channels. However, the manner in which large strands of dsDNA are condensed in such a lipid complex are far from trivial. Therefore simply extending our findings to the transfection of plasmid sized complexes is something we do not feel comfortable doing.